# A Effectiveness-and Efficiency-Based Improved Approach for Measuring Ecological Well-Being Performance in China

**DOI:** 10.3390/ijerph20032024

**Published:** 2023-01-22

**Authors:** Bei He, Xiaoyun Du, Junkang Li, Dan Chen

**Affiliations:** 1School of Engineering Management and Real Estate, Henan University of Economics and Law, Zhengzhou 450000, China; 2School of Management, Center for Energy, Environment & Economy Research, Zhengzhou University, Zhengzhou 450000, China

**Keywords:** ecological well-being performance (EWP), EWP effectiveness, EWP efficiency, China, sustainable development

## Abstract

Finding solutions to the challenges posed by China’s urbanization is an urgent, pressing global concern. An effective approach for evaluating the ecological well-being performance (EWP) is a guideline for improvement. Most previous studies have focused on the evaluation of EWP efficiency without considering the effectiveness of the EWP, which may mislead the practice of improving the EWP. This paper proposed a bi-dimensional effectiveness and efficiency perspective evaluation of the EWP for pursuing sustainable development goals. The Ecological Consumption Index and the Human Development Index are selected to evaluate indicators for the EWP. The entropy method, line-weighted method, and four-quadrant evaluation framework are used to disclose EWP effectiveness. A Super SBM model and the DEA moving split-windows analysis method are applied to calculate the EWP efficiency. Data from 30 provinces in China for the period of 1997 to 2019 have been collected for empirical study to demonstrate the effectiveness of the proposed method. The main findings of the case study are: (1) The ECI and HDI increased during the study period, while the annual average value of the EWP efficiency among 30 provinces in China has decreased with fluctuation; (2) provinces in southern China and Chongqing have a low level of ECI and demonstrate good performance in the HDI; and (3) most developed regions, such as Beijing, Shanghai, and Guangdong, have not presented the best EWPs. The results of this study can provide a basis for understanding the EWP in China so as to formulate targeted sustainable-development strategies.

## 1. Introduction

Over the course of the last two decades, the world has witnessed an unparalleled urbanization. According to the 2018 Revision of World Urbanization Prospects [1], the global urbanization rate reached 55.3% in 2018. It is predicted that, during the period of 2018 to 2050, the urbanization rate in the developed regions of the world will slowly rise from 78.7% to 86.6%, while in less-developed regions the urbanization rate will rise from 50.6% to 65.6%. This demonstrates that the majority of global urbanization will occur in developing nations over the next 30 years, primarily in Asian and African regions. Among these regions, China is recognized as a significant representation of developing countries due to its size and influence. Since the Chinese government decided to implement economic reform and opened up in 1978, China has experienced an unprecedented process of urbanization. According to the National Plans for New-type Urbanization (2014–2020) [2] and the China Statistical Yearbook [3], the urban population in China has increased dramatically from 170 million in 1978 to 901.99 million in 2020; the population urbanization rate of China has grown from 17.92% to 63.9% during that period. Therefore, it is very meaningful to study urbanization in China, which can bring various benefits to the global sustainable development goals.

Cities are the major engine for the economic development of a country and have undoubtedly become a vital component of human society. However, as high-density human settlements, cities have been encountering the most acute conflicts between human activities and the ecological environment. A number of serious problems and significant difficulties have emerged due to the deterioration of the urban environment and the over-exploitation of natural resources [4]. According to the official statistics compiled by the Yearbook of the United Nations [5], 74.3% of the energy consumption and 75.5% of pollutant emissions around the world are from cities. As for China, the process of large-scale urbanization has caused various types of urban environmental problems to arise, such as sandstorms, air pollution, the disruption of the ecological balance, a shortage of ecological resources, and the greenhouse gas effect. For example, extreme weather caused by the greenhouse gas effect in Zhengzhou, central China, has caused more than CNY 400,000 in car damage and CNY 6.25 billion in financial losses [6]. It is commonly appreciated that the pressure of environmental pollution in China will continue to be significant in the ongoing process of urbanization. Therefore, finding solutions to the challenges posed by China’s urbanization is urgent and has generated pressing global concerns.

Given that the ecological environment is facing increasing challenges due to the continuous global urbanization process, it is extremely important to strike a better balance between economic growth and environmental protection, which can be reflected in the concept of eco–environmental performance. According to the National Bureau of Statistics of China [3], overall investments in environmental protection by the central government of China have increased dramatically, rising from CNY 99.582 billion in 2007 to CNY 633.34 billion in 2020. Local governments in China have also implemented a series of policies to improve eco–environmental performance. For instance, the government of Shanghai [7] implemented a three-year act plan for environmental protection and construction (2021–2023). However, ecological and environmental performance in China is poor, as has been demonstrated by previous studies. According to the World’s Environmental Performance Index (EPI): 2020 Report [8], which is conducted jointly by the Yale University Center for Environmental Law and Policy, the Socioeconomic Data and Applications Center (SEDAC) at Columbia University, and the World Economic Forum, China only received a final EPI performance score of 65.1 out of a possible 100 points and was ranked 109th among the total of 180 evaluated countries in the world. According to the Global Energy Architecture Performance Index Report 2016 (EAPI) issued by the World Economic Forum, China scored 89th out of the total of 126 countries in the world that were examined for their energy architecture [9]. In order to improve eco–environmental performance, both the central and local governments of China have attached great importance to improving environmental performance. For example, the Government Work Report of the 17th National Congress of the Communist Party of China suggested achieving ecological civilization and enhancing environmental quality were the primary objectives for resolving significant environmental concerns in 2007. At the 18th National Congress of the Communist Party in 2012, an ambitious plan titled “Ecological Civilization” was unveiled and advocated for, which was essential for enhancing China’s sustainable development. It is therefore important to evaluate ecological performance in China systematically and to explore the effectiveness of the proposed measures. Systematic evaluation is the guideline for improving the ecological environment performance of China and can provide a reference to balance the relationship between China’s economic growth and environmental protection.

In order to disclose ecological–environmental performance, a greater focus has been placed on ecological efficiency from the perspective of economic benefits, using the theory of “Pareto optimality” as the evaluated foundation [10,11,12]. There has recently been a progressive shift toward placing a greater emphasis on human well-being in the pursuit of urban sustainability: this approach seeks to improve the contentment and comfort of the living environment for urban people. Therefore, it is imperative to properly understand ecological efficiency from the perspective of human well-being. This is considered to be an extension of sustainable urbanization and green development. Accordingly, ecological well-being performance (EWP), which is an upgraded version of ecological efficiency, was initially proposed by Daly [13]. EWP is an important concept that measures the transformation of the minimization of ecological-impact inputs into the maximization of human-well-being outputs. The evaluation of the EWP has the potential to break through the limitations of traditional GDP benefits by evaluating the quality and performance of human well-being. Therefore, the EWP has been suggested and regarded as a quantitative measurement of the harmony extant between the natural environment and human well-being.

An effective and proper approach should be developed and employed in order to evaluate the ecological well-being performance (EWP). In other words, the current effectiveness of various ecological resource inputs and output can be understood only when the EWP is evaluated in an appropriate and efficient manner. However, the process of rapid urbanization cannot be diagnosed in the absence of a comprehensive evaluation method such as the EWP [4]. As a result, administrative institutions are unable to devise appropriate actions and effective measures to address problem areas. Through a critical review of the previous research on the EWP, it is apparent that most previous studies focused on the efficiency of the EWP without considering the effectiveness of the EWP; this may mislead the practice of improving the EWP. According to Du [6], effectiveness is the basis for success, while efficiency is the minimum condition for sustaining success after it has been achieved; efficiency is about how to do things right, while effectiveness is about doing the right things. Therefore, evaluating the EWP from a single perspective of effectiveness or efficiency may produce a biased perception. In other words, a bi-dimensional effectiveness and efficiency perspective evaluation of the EWP is proposed to curb sustainable performance.

Having an appreciation for the limitations of the previous research has led to the formulation of the following research question for this study: how can the ecological well-being performance (EWP) be effectively evaluated from a holistic perspective that takes into consideration the effectiveness and efficiency of the EWP? The effectiveness of the EWP reflects the status performance of EWP inputs and outputs, while the efficiency of the EWP measures how input factors contribute to output factors. In this study, the amount of ecological consumption is regarded as the input factor of EWP, and the level of residential welfare is provided as an EWP output factor. It is vital to take into consideration both the ecological environment’s input as well as the resident welfare output in order to improve the development status of ecological welfare in a particular place. The EWP efficiency centers on how EWP inputs are transformed into social welfare outputs, which can reveal the practical problem of how human beings can achieve happiness through more intensive production and lifestyle choices. Therefore, the purpose of this research is to develop a novel evaluation approach to assessing the ecological well-being performance (EWP) and to examine relevant policy measures and the appropriate consequences for promoting residential happiness and ecological protection. In order to achieve the ambitious objectives of an ecological civilization and sustainable development, the research findings will serve as references for decision-makers and urban managers working in the government agencies responsible for ecological protection and urban construction. 

In accordance with this overarching research aim, the following are four specific research objectives: (1) to establish the proper index framework for evaluating the EWP (Section Three); (2) to develop a novel evaluation model of the EWP (Section Four); (3) to demonstrate the effectiveness of the established EWP evaluation model by a Chinese, context-based case study (Sections Five and six); and (4) to draw out policy implications for urban planning and governance by synthesizing the research findings (Section Seven).

## 2. Literature Review

In Section 2.1, this paper will review existing studies on the concept of ecological well-being performance. The efficiency evaluation methods in previous studies were mainly divided into two groups based on the number of input and output factors used in the efficiency evaluation: efficiency measurement for a single factor and efficiency measurement for total factors, which will be summarized in Section 2.2 and Section 2.3.

### 2.1. The Concept of EWP

The study of the EWP was first prompted in 1974 by Herman E. Daly’s comparative analysis of the state of sustainable development in a number of nations, which served as the impetus for the research [14]. The ratio of the welfare utility of ecosystem services to the natural-resource consumption was conceived as “Service/Throughout” (the ratio of service to flux) to illustrate the per-unit welfare value created by the utilization of ecological resources. Schaltergger and Burritt [15] introduced the concept of ecological efficiency, which is characterized by the economic value produced by the ecological environment in relation to economic development and environmental impact. 

Academic researchers have always regarded economic output as a welfare-substituting variable and, over the past few years, have conducted a great number of studies focused on the issue of “ecological efficiency.” Ecological efficiency was primarily discussed from the perspectives of product, organization, region, and nation. Academic researchers primarily examined the concept of ecological efficiency and its meaning from the various perspectives of products, organizations, regions, and countries. (1) According to a study by the World Business Council for Sustainable Development, ecological efficiency was defined as “the requirement that goods and services in their full life cycle can meet human needs, have a price advantage, have less environmental impact, and do not exceed the ecological carrying capacity of the earth” [16]. According to report findings published by the Industry of Canada, ecological efficiency can be defined as “the consolidation of reducing production costs and maximizing the advantages of commodities”. The Atlantic Canada Opportunities Agency (ACOC) noted in its report that “ecological efficiency can provide high-quality products and services while maintaining a focus on the environmental impact of the production process, to minimize environmental damage, and to reduce resource consumption” [17]. (2) The Organization for Economic Co-operation and Development (OECD) described the concept of “ecological efficiency” as “the measurement of the link between input and output in the field of ecological environment”, namely, “the use of the fewest resources and the shortest amount of environmental input to generate the most economic value”, from an organizational perspective [18]. Schaltegger and Burritt [15] defined eco-efficiency as one of the crucial aspects for enterprise environmental management; it can be measured as “economic performance/environmental performance”. (3) From the perspectives of regions and nations, the European Environment Agency (EEA) defined ecological efficiency as “the economic benefit created by a unit of natural resources after taking into account the economic, social, environmental, and other characteristics of a country or region” [19,20]. This index can be used to determine the extent to which a country or region has attained sustainable development.

### 2.2. Efficiency Measurement from a Single Factor

Typically, efficiency measurement for a single factor usually only considers the unit output capacity of the factor. Researchers mostly use the value output per unit of ecological input as the evaluation method of eco–environmental efficiency, often using the economic output as the final value output. For example, the World Business Council for Sustainable Development (WBCSD) adopted the formula of “ecological efficiency = Value of products or services/Environmental impact” [21]. The formula proposed by Statistics of Finland is “Ecological efficiency = Improvement in quality of life/(Consumption of natural resources + Environmental loss + Economic cost)”. The calculation formula proposed by Schaltegger and Burritt [15] is “Ecological efficiency = Output/Increase in environmental impact”. At the same time, when scholars use the ratio evaluation method to measure ecological efficiency, the calculation formula does not strictly adopt the single form of “input/output”, but also uses “output/input” to represent efficiency. For example, the calculation formula proposed by Muller and Sterm [22] is “Ecological efficiency = Environmental impact/Output value”. In 2003, the United Nations Conference on Trade and Development (UNCTAD) proposed that the formula for measuring eco-efficiency is “Ecological efficiency = Value/Environmental impact” [23]. In this formula, “value” mainly refers to output indicators, such as product or service value and product sales, while “environmental impact” is represented by ecological environment indicators such as energy consumption, water consumption, CO_2_ emissions, and ozone-layer-depletion emissions. The calculated framework of efficiency measurement for the single-factor method is straightforward, and it is simple to gather and calculate data. It is commonly employed in the earliest stages of ecological efficiency evaluation [24,25]. This calculating approach ignores the substitution effect of other production elements such as labor, capital, machinery, and equipment, rendering the findings of the computation largely one-sided [26].

### 2.3. Efficiency Measurement from Total Factors

As the research on the evaluation of ecological—environmental efficiency has progressed, an increasing number of researchers have adopted the method of efficiency measurement for total factor due to substantial flaws in the efficiency measurement of a single factor. The concept underlying the efficiency measurement for total factor is as follows: first, the efficiency production frontier is determined based on the inputs and outputs of the evaluated unit. The efficiency measurement value of the evaluated unit is then determined by comparing the distance between the evaluated unit and the efficiency frontier. 

According to the stated variations in the construction methods of production (efficiency) frontier by Farell in 1957, it can be separated into two calculated methods: parametric and non-parametric [27]. By adjusting the production (efficiency) function, the parametric method is mostly utilized to fit. The stochastic frontier analysis (SFA), thick frontier approach (TFA), and the distribution free method (DFA) are commonly utilized. For instance, Reinhard et al. [28] evaluated the environmental efficiency of Dutch ranchland by using the SFA and DEA methods. The results of the calculations demonstrated that the SFA permits the testing of hypotheses; however, the monotonicity hypothesis is denied. The DEA approach can be used to assess the efficiency of each item (technical efficiency, scale efficiency, comprehensive efficiency, etc.) in order to analyze the benefits and drawbacks of the two aforementioned methods of measuring efficiency.

The non-parametric method differs from the parametric method in that it does not require the setting and testing of a fixed model. Instead, it primarily employs the piecewise convex function approximation method to estimate the efficiency frontier, which is represented primarily by data envelopment analysis (DEA). Murty et al. [29] measured the environmental efficiency of the Indian sugar industry using the DEA approach and conducted an empirical investigation of the impact of pollution discharge from different water bodies on the sector’s environmental efficiency. Based on the “input-output” concept of the DEA model, Li & Hu [11] calculated the ecological total-factor energy efficiency (ETFEE) of 30 provinces in China from 2005 to 2009, using the SBM model while taking into account the unwanted output of pollutants such as CO_2_ and SO_2_. During the study period, the ETFEE efficiency value of all Chinese provinces remained close to 0.6, which is a low level, and there were considerable variances between regions. Azadeh et al. [30] employed the DEA model, principal component analysis, numerical classification (NT), and other techniques to evaluate the total-factor energy efficiency of energy-intensive manufacturing industries, using the refining industry of a few OECD nations as a case study. Wei et al. [31] measured the energy efficiency of China’s steel industry at the provincial level from 1994 to 2013 using the DEA method, and further decomposed the effect of energy efficiency using the Malmquist index, namely, the “Production frontier shifting effect” and “Catching up effect” caused by technological advancement and changes in technical efficiency, respectively. Using the DEA model, Chien and Hu [32] analyzed the effect of renewable energy on technical efficiency in 45 countries from 2001 to 2002. Zhang et al. [33] measured the eco-efficiency performance of industrial systems in various provinces of China using the DEA approach, and their findings indicated that Tianjin, Shanghai, Guangdong, Beijing, Hainan, Qinghai, and other provinces had elevated eco-efficiency levels throughout the study period.

Numerous research studies are currently devoted to the empirical analyses of EWPs at the national, provincial, urban, and municipal levels. Zhang et al. (2018) [34] investigated the EWP of 82 countries in 2012 and found that industrialized nations and G20 nations have relatively low EWP values. Fang et al. [35] analyzed the changing trend of the EWP of 30 provinces in China from 2005 to 2010, using data from 30 provinces. Using exploratory spatial-data analysis, Xu et al. [36] examined the spatiotemporal characteristics of the EWP across 30 Chinese provinces. Long and Wang [37] provided a comprehensive evaluation of Shanghai’s EWP from 2006 to 2014. Hickel [38] conducted a comprehensive investigation of the EWP of 278 Chinese cities between 2005 and 2016 using the DEA approach. Xiao and Zhang [39] estimated the EWP at the provincial level in China for 2004–2015 using an improved SFA model. DiMaria [40] assessed the EWP at the national level of China using an improved DEA model. Bian et al. [41] used the SE-SBM model to measure the EWP of 30 provincial capital cities in China from 2011 to 2016. Zhong et al. [42] explored the multi-dimensional EWP of 283 cities in China from 2003 to 2018 using the coupling coordination model. Zhang et al. [43] evaluated the EWP on the belt and road regions from 1990 to 2017 according to the consumption–pressure–output–efficiency method. Xia and Li [44] used a two-stage DEA model and the Malmquist index to evaluate the EWP and spatial correlation analysis of the Beijing–Tianjin–Hebei urban agglomeration from 2006 to 2019.

To summarize the existing literature, there are some gaps in the extant research. (1) In terms of research content, previous studies on ecological well-being performance mainly adopt the method of efficiency measurement for evaluation. There is a lack of evaluation from the effectiveness perspective. Effectiveness is the basis for success, while efficiency is the minimum condition for sustaining success after it has been achieved. Therefore, this study examines the evaluation of the EWP from a comprehensive view of both effectiveness and efficiency. Meanwhile, the same indicator system is adopted for both the effectiveness evaluation and efficiency evaluation of the EWP. Thus, the evaluation results for EWP effectiveness and EWP efficiency can be compared horizontally. (2) In terms of the research methods, existing studies on EWP rely predominantly on the static DEA model. The static DEA model can only evaluate the relative efficiency of multiple DMUs over the same time period, which renders the evaluation results non-comparable across time. The DEA moving split-windows analysis is a proven method for evaluating dynamic DEAs when working with panel data. Therefore, this study makes an attempt to adopt the DEA moving split-windows analysis method to conduct empirical research on panel data at a regional level in China over an extended period of 22 years.

## 3. Methodology

According to the information provided in Section 1, the purpose of this study is to establish an innovative and comprehensive approach for evaluating the performance of ecological well-being. To achieve this overarching research aim, a research flowchart was designed, (shown in Figure 1). 

As is shown in Figure 1, the first step was to establish an evaluation index system for the EWP. In this study, the framework of indicators was constructed based on the input–output model; the effect evaluation index system of the EWP effectiveness was established based on the Ecological Consumption Index and the Human Development Index, and the evaluation index system of the EWP efficiency was established based on input factors and output factors. Second, it was necessary to establish an evaluation model for the EWP. In this study, the effect measurement model of the EWP effectiveness was established based on entropy weight, the linear-weighting method, and the quartile method, and the efficiency of the EWP was calculated based on the Super-SBM model and the moving split-windows DEA model. Finally, by collecting the time data of 30 sample provinces in China from 1997 to 2019, an empirical study was carried out to verify the effectiveness of the designed EWP evaluation model. Specifically, the effectiveness of the model was verified by analyzing the evolution of the EWP effectiveness and spatiotemporal variation and the spatial differences of the EWP efficiency.

## 4. Evaluation Indicators for Measuring the Ecological Well-Being Performance (EWP)

### 4.1. Overall Framework for EWP Indicator Selection

An index system for measuring the ecological well-being performance (EWP) as established and applied by introducing an input–output model (IOM) with a reference to the “End-Means” analysis framework. According to previous studies by Daly [45] and Costanza [46], the “End-Means” analysis framework was developed in order to define the relationship between ecological input and well-being output from a new point of view. The “End-Means” analysis framework places an emphasis on the process of transitioning from ultimate means to ultimate purposes. Consequently, ultimate efficiency can be regarded as follows:

Specifically, ultimate means are defined as natural consumption and evaluated by ecological consumption. Ultimate ends are defined as improvement in well-being and utility satisfaction and are evaluated by human well-being. Therefore, by utilizing the “End-Means” framework and the equation presented above (see Figure 2), an input–output model may be used to appropriately apply index selection and a comprehensive evaluation of the EWP.

By referring to the “End-Means” analysis framework addressed in Section 2, this section will present and employ a model named the input–output model (IOM) for establishing a set of indexes to evaluate the ecological well-being performance. In line with the principles of the input–output model, two types of indicators are needed for determining the values of EWP indexes to evaluate the effectiveness and efficiency of the EWP; namely, input indicators and output indicators. Accordingly, the index value for reflecting the effectiveness of the EWP in referring to a specific region is expressed by the ecological consumption (EC) and human development (HD). Meanwhile, using EC as an input factor and HD as an output factor will contribute to the index value, showing the efficiency of the EWP. 

Therefore, there are three steps to establishing the EWP evaluation indexes: (a) to identify and select indicators of ecological consumption for the EWP effectiveness evaluation; (b) to identify and select indicators of human development against each selected indicator of ecological consumption for the EWP effectiveness evaluation; and (c) to establish an indicator system for measuring the EWP efficiency based on the selected indicators of EC and HD. 

### 4.2. Indicators for EWP Effectiveness Assessment 

#### 4.2.1. Indicator Selection of Ecological Consumption Index 

Ecological consumption can be categorized primarily along two dimensions: resource consumption and environmental pollution. As is indicated in Table 1, a variety of appropriate indicators have been reviewed in previous studies which focused on assessing and discussing the performance of natural resource consumption and environmental pollutant emissions.

The criteria for the selection of appropriate indicators proposed by Cepoi and Toma [63] were adopted to select indicators. The selection primarily consisted of nine evaluation criteria: (1) feasibility, (2) rationality, (3) data availability, (4) number limitation, (5) completeness, (6) representativeness, (7) relevance, (8) measurability, and (9) compatibility [64]. Typically, energy, land, and water are the three primary components that represent natural resource consumption and usage. The typical indicators of natural resource consumption have been categorized into three groups based on the relevant studies and data availability: energy consumption, land consumption, and water consumption. The indicators of typical environmental pollutant emissions were also separated into three categories: wastewater discharge, exhaust emissions, and municipal solid-waste discharge. The vast majority of previous studies utilized the total emissions of “three wastes” (wastewater, municipal solid waste, and exhaust gas) to evaluate the performance of environmental pollutant emissions, particularly in the industrial sector. However, indicators of the “three wastes” in the industrial sector have only displayed the effects of industry on the natural environment in a particular region and cannot completely depict the relationship between regional overall growth and ecological environment. In this study, the total emissions of the “three wastes” were chosen as evaluation indicators, encompassing industrial and municipal pollutant resources.

Consequently, candidate indicators for the ECI (as shown in Table 1) are filtered and selected based on the application of certain indicator selection criteria. Specifically, energy resource consumption is evaluated by the standard coal consumption per capita (ECI-X1); land resource consumption is evaluated by the built-up area per capita (ECI-X2); water resource consumption is evaluated by the water supply per capita (ECI-X3); wastewater discharge is assessed via the COD of wastewater per capita (ECI-X4); exhaust gas emissions are evaluated by the SO_2_ emissions per capita (ECI-X5) and the fumes emissions per capita (ECI-X6). Municipal solid-waste discharge is assessed via the quantity of domestic-garbage-clearance disposal (ECI-X7) and industrial solid-waste discharge per capita (ECI-X8). 

#### 4.2.2. Indicator Selection of Human Well-Being Index

In extant research, gross domestic product (GDP) and gross national product (GNP) are the most preferred methods for measuring well-being performance [65,66]. However, these economic indicators have significant limitations when used to assess the actual degree of human welfare [66,67]. Therefore, multiple indices have been developed to analyze the external environmental circumstances of residential welfare in order to quantify residential objective happiness [67,68,69,70,71]. Table 2 provides a brief summary of typical indictors for evaluating well-being performance.

The Human Development Index (HDI), which was proposed by the United Nations Development Programme (UNDP) in 1990, has steadily achieved international prominence and is currently regarded as one of the most well-known and authoritative indicators of residential objective welfare for two primary reasons: (1) the HDI’s theoretical basis can provide an overall evaluation of human objective happiness in terms of freedoms, opportunities, and capabilities and (2) the HDI index can be compared between different cities, regions, and even countries due to its authority and data availability throughout the world [80,81]. Consequently, the Human Development Index (HDI) was selected to quantify objective well-being for the EWP effectiveness evaluation in this study. Using the evaluation criteria of the HDI, this study has defined and measured the well-being output in three primary dimensions: economic development, residential health, and residential education level. In particular, economic development is measured by the GDP per capita (HDI-Y1), residential health level is measured by life expectancy at birth (HDI-Y2), and residential education level is measured by the audit literacy rate (HDI-Y3) and the comprehensive education enrollment rate (HDI-Y4) [82].

### 4.3. Indicators for EWP Efficiency Assessment

According to the ECI index model and the HDI index described in Section 4.2, an indicator evaluation system for measuring the EWP has therefore been established. The main features of the indicators were demonstrated, and are shown in Table 3.

## 5. Measurements of the Improved EWP Model

### 5.1. Measurements for EWP Effectiveness Assessment

#### 5.1.1. Ecological Consumption Index (ECI)

In this study, the evaluation of the Environmental Consumption Index (ECI) mainly includes six aspects: energy resource consumption (ECI*_er_*), land resource consumption (ECI*_lr_*), water resource consumption (ECI*_wr_*), wastewater discharge (ECI*_ww_*), exhaust gas emission (ECI*_eg_*), and municipal solid-waste discharge (ECI*_msw_*). The Environmental Consumption Index (ECI) is calculated based on the following model:(1)$$ECI=ECI_{er}+ECI_{lr}+ECI_{wr}+ECI_{ww}+ECI_{eg}+ECI_{msw}$$

The six dimensions of the Environmental Consumption Index (ECI) can be defined as follows:(2)$$ECI_{er}=\sum_{j=1}^{n_{ECI_{er}}}w_{j\left(ECI_{er}\right)}\cdot P_{ij\left(ECI_{er}\right)}\;j=1,2,3,\cdots ,n_{ECI_{er}}$$
(3)$$ECI_{lr}=\sum_{j=1}^{n_{ECI_{lr}}}w_{j\left(ECI_{lr}\right)}\cdot P_{ij\left(ECI_{lr}\right)}j=1,2,3,\cdots ,n_{ECI_{lr}}$$
(4)$$ECI_{wr}=\sum_{j=1}^{n_{ECI_{wr}}}w_{j\left(ECI_{wr}\right)}\cdot P_{ij\left(ECI_{wr}\right)}j=1,2,3,\cdots ,n_{ECI_{wr}}$$
(5)$$ECI_{ww}=\sum_{j=1}^{n_{ECI_{ww}}}w_{j\left(ECI_{ww}\right)}\cdot P_{ij\left(ECI_{ww}\right)}j=1,2,3,\cdots ,n_{ECI_{ww}}$$
(6)$$ECI_{eg}=\sum_{j=1}^{n_{ECI_{eg}}}w_{j\left(ECI_{eg}\right)}\cdot P_{ij\left(ECI_{eg}\right)}j=1,2,3,\cdots ,n_{ECI_{eg}}$$
(7)$$ECI_{msw}=\sum_{j=1}^{n_{ECI_{msw}}}w_{j\left(ECI_{msw}\right)}\cdot P_{ij\left(ECI_{msw}\right)}j=1,2,3,\cdots ,n_{ECI_{msw}}$$

$\mathrm{The}\;\mathrm{variables}\;P_{ij\left(ECI_{er}\right)}$, $P_{ij\left(ECI_{lr}\right)}$, $P_{ij\left(ECI_{wr}\right)}$, $P_{ij\left(ECI_{ww}\right)}$, $P_{ij\left(ECI_{eg}\right)}$, and $P_{ij\left(ECI_{msw}\right)}$ represent the normalization values of the indicator *j* in year *i* across six dimensions: energy resource consumption, land resource consumption, water resource consumption, wastewater discharge, exhaust gas emission, and municipal solid-waste discharge, respectively, while $n_{ECI_{er}}$, $n_{ECI_{lr}}$, $n_{ECI_{wr}}$, $n_{ECI_{ww}}$, $n_{ECI_{eg}}$, and $n_{ECI_{msw}}$ represent the number of indicators which are selected for conducting an evaluation of the ECI performance from the above six dimensions. Finally, $w_{j\left(ECI_{er}\right)}$, $w_{j\left(ECI_{lr}\right)}$, $w_{j\left(ECI_{er}\right)}$, $w_{j\left(ECI_{er}\right)}$, $w_{j\left(ECI_{er}\right)}$, and $w_{j\left(ECI_{er}\right)}$ denote the weighting values of the indicator j in view of the above six dimensions, respectively.

It is essential and crucial to weight values between different indicators by evaluating performance and calculating the index for ecological consumption. There are two primary approaches for determining indicator weighting values: the subjective weighting method and the objective weighting method. Specifically, the subjective weighting methods mainly include the Delphi method, analytic hierarchy process (AHP), binomial coefficient method, minimum scoring method, and the sequential scoring method. The objective weighting methods mainly include the principal component analysis, entropy method, deviation and mean square deviation method, and the multi-objective programming method. Among these methods, the objective weighting methods are regarded as effective evaluation methods for establishing weightings between various indicators [83,84]. In this study, the entropy method, which is widely regarded for determining objective weighting values, was chosen to establish indicator weightings and conduct indicator evaluations. The entropy method is typically employed and engaged with five procedures, as follows:

(a) Indicator normalization

Indicator normalization is adopted to eliminate effects for different dimensions and magnitudes of different indicators for performance evaluation.

The normalized value of positive indicators, $P_{ij},\;$can be calculated as follows:(8)$$P_{ij}=\frac{v_{ij}-min\left(v_{j}\right)}{max\left(v_{j}\right)-min\left(v_{j}\right)}$$

The normalized value of negative indicators, $P_{ij}$, of negative indicators can be calculated as follows:(9)$$P_{ij}=\frac{max\left(v_{j}\right)-v_{ij}}{max\left(v_{j}\right)-min\left(v_{j}\right)}$$
$\mathrm{where}\;P_{ij}$ denotes the normalized value of the indicator j in the year i for the surveyed period of m years, $V_{ij}\;$demotes the original value of the indicator j in the year i in the surveyed period, and Max($V_{j}$) and Min($V_{j}$) represent the maximum value and minimum value of $V_{ij}$, respectively.

$\mathrm{The}\;\mathrm{variable}\;f_{ij}\;$represents the standardized value of the normalized indicator, j, in the year i, which can be calculated as follows:(10)$$f_{ij}=\frac{P_{ij}}{\sum_{i=1}^{n}P_{ij}}$$

(b) Assessment of indicator entropy value

$\mathrm{The}\;\mathrm{variable}\;e_{j}$ denotes the entropy value of the normalized indicator, *j*, for the evaluated period of m years by applying the entropy theory, in which $k=\frac{1}{lnm}$ and 0 ≤ $e_{j}$ ≤ 1.
(11)$$e_{j}=-k\overset{m}{\underset{i=1}{\sum }}f_{ij}\cdot lnf_{ij}$$

(c) Coefficient calculation of difference

$\mathrm{The}\;\mathrm{variable}\;g_{j}$ represents the coefficient of difference for the indicator *j*, which can be calculated as follows:(12)$$g_{j}=1-e_{j}$$

(d) Establishment of weighting values

$\mathrm{The}\;\mathrm{variable}\;W_{j}$ denotes the weighting values of the indicator *j* by applying the entropy method, which can be defined as:(13)$$w_{j}=\frac{g_{j}}{\sum_{j=1}^{n}g_{j}}$$

#### 5.1.2. Human Development Index (HDI)

The Human Development Index (HDI) mainly includes three aspects: the income index (II), health index (HI), and the education index (EI). The equation for the HDI has been described as follows:(14)$$HDI=\sqrt[{3}]{II\times HI\times EI}$$

Specifically, the II, HI, and EI can be calculated by applying the three following equations using the index benchmark of the HDI from the “Human Development Report” authorized by UNDP, shown in Table 4.
(15)$$II=\frac{\log\left(Per\;capital\;GDP\right)-\log100}{\log40000-\log100}$$
(16)$$HI=\frac{Life\;expectancy\;at\;birth-25}{85-25}$$
(17)$$EI=\frac{1}{3}\times Overall\;enrollment\;rate+\frac{2}{3}\times Adult\;literacy\;rate$$

#### 5.1.3. Four-Quadrant Evaluation Framework

A two-axis, four-quadrant scatter plot was established in order to explain the correlational relationship between the input of ecological consumption and the output of human well-being. The effectiveness assessment for the EWP can be explained in four types according to the average values of the ECI and HDI, namely, Type I—high ECI–high HDI (H-H); Type II—high ECI–low HDI (H-L); Type III—low ECI–low HDI (L-L); and Type IV—low ECI–high HDI (L-H), shown in Table 5. Specifically, Type I (H-H) and Type III (L-L) represent that the EPW effectiveness is under sustainable development, Type II (H-L) represents the unsustainable development of the EWP effectiveness, and Type IV (L-H) represents the sustainable development of the EWP effectiveness. Therefore, a four-quadrant evaluation framework for an effectiveness assessment of the EWP can be established based on the above types of classification, as is shown in Figure 3.

### 5.2. Measurements for EWP Efficiency

#### 5.2.1. Super-SBM Method

The most widely used methods for efficiency measurement and calculation are the single input–output ratio, the comprehensive assessment of the indicator system, the stochastic frontier analysis (SFA), and the data envelopment analysis (DEA) [85,86]. However, because the EWP is dependent on a number of different dimensions, the first two approaches described above for conducting an evaluation of the EWP have a number of significant drawbacks. The SFA and DEA methods have been widely adopted to conduct quantitative efficiency analyses that take into account multiple inputs and multiple outputs. In particular, as a non-parametric analysis method, DEA does not require an excessively large number of evaluation indicators, model specifications, or parametric tests when it is applied in conducting an efficiency assessment. The CCR model, which was first proposed by Charnes et al. in 1978 [87], was used to measure the relative efficiency of individual decision-making units (DMUs) with the same inputs and outputs based on a series of assumptions, including perfect competition and constant returns to scale, as the first DEA model. The slack-based measure model (SBM), which was proposed as a typical, non-radial DEA model by Tone in 2001 [88], has directly evaluated the inefficiency degree of DMUs by making use of the mean slackness of different inputs and outputs. When utilizing the traditional SBM model, the efficiency value of all DMUs will be constrained to fall within a range from 0 to 1, regardless of their specific design.

In most cases, there will generally be several different DMUs for performance evaluation, and the efficiency values will appear to be equal to 1 at the same time. Under such a circumstance, however, it is difficult to carry out more comparisons between different SBM-efficient DMUs. Tone [89] proposed the Super-SBM model as a means of improving the traditional SBM model and addressing the deficiencies caused by the assessment limitations. When the upgraded SBM model is used, the SBM efficiency of DMUs can be compared without the range limitation of [0, 1], which means that the efficiency value can be greater than 1. As a consequence, the Super-SBM model was utilized for the efficiency assessment of the EWP in this study. The following measures and steps were taken to accomplish this:

Assuming that there are *n* DMUs participating in the performance evaluation, the input matrix and the output matrix can be formed individually, as follows:(18)$$X=\left(x_{ij}\right)\in R^{m\times n}>0,Y=\left(y_{ij}\right)\in R^{q\times n}>0$$

The fractional equation of the non-angular SBM model can be described as follows:(19)$$min\delta =\frac{1-\frac{1}{m}\sum_{i=1}^{m}\frac{s_{i}^{-}}{x_{ik}}}{1+\frac{1}{q}\sum_{r=1}^{q}\frac{s_{r}^{+}}{y_{rk}}}\;X=\left(x_{ij}\right)\in R^{m\times n}>0,\;\mathrm{Subject}\;\mathrm{to}\{\begin{array}{c} \overset{n}{\underset{j=1,j\neq k}{\sum }}x_{ij}\lambda_{j}+s_{i}^{-}=x_{ik}\\ \overset{n}{\underset{j=1,j\neq k}{\sum }}y_{rj}\lambda_{j}-s_{r}^{+}=y_{rk}\\ \lambda ,s^{+},s^{-}\geq 0\\ i=1,2,\cdots ,m;r=1,2,\cdots ,q;j=1,2,\cdots ,n \end{array}$$
where δ is the efficiency value, 0 < δ ≤ 1; $x_{ik}$ and $y_{rk}$ represent the input variables and outputs variables of DMUs, respectively, m and q are the number of input and output indicators, respectively, $s_{i}^{-}$ and $s_{r}^{+}$ represent the slack variables of input and output, respectively, and λ represents the weight vector.

The Super-SBM model is an improvement of the SBM model, which is provided as follows:(20)$$\delta^{*}=min\delta =\frac{1+\frac{1}{m}\sum_{i=1}^{m}\frac{s_{i}^{-}}{x_{ik}}}{1-\frac{1}{q}\sum_{r=1}^{q}\frac{s_{r}^{+}}{y_{rk}}}\;\mathrm{Subject}\;\mathrm{to}\{\begin{array}{c} \overset{n}{\underset{j=1,j\neq k}{\sum }}x_{ij}\lambda_{j}-s_{i}^{-}\leq x_{ik}\\ \overset{n}{\underset{j=1,j\neq k}{\sum }}y_{rj}\lambda_{j}+s_{r}^{+}\geq y_{rk}\\ \lambda ,s^{+},s^{-}\geq 0\\ i=1,2,\cdots ,m;r=1,2,\cdots ,q;j=1,2,\cdots ,n;j\neq k \end{array}$$
$\mathrm{where}\;\delta^{*}$ is the score of the efficiency value for the EWP. The evaluated DMU is relatively effective when $\delta^{*}$ ≥ 1. However, it is relatively invalid for the evaluated DMU when ≤ 1. The higher the value $\delta^{*}$ is, the better the efficiency of the EWP is.

#### 5.2.2. DEA moving split-windows analysis

The static DEA model can be used to assess the relative efficiency of the EWP for various DMUs within the same period. It is incomparable to evaluate efficiency of the EWP across different samples and years. The Malmquist index approach and the DEA moving split-windows analysis are the two most important evaluation approaches for dynamic DEA when working with panel data. The Malmquist index approach was first deployed and utilized to evaluate the efficiency of production technology in 1992 [90]. The DEA moving split-windows analysis was initially developed in 1985 and was used to evaluate the effectiveness of the maintenance agency for American Airlines [91]. The DEA moving split-windows analysis was widely utilized for the efficiency evaluation of the ecological environment [57,92].

By utilizing the DEA moving split-windows analysis method, it is possible to make repeated use of DMUs for an efficiency evaluation. This makes it possible to increase the total number of DMUs that are reviewed, as well as to improve the reliability and stability of the findings of such evaluations. The moving average approach was introduced and presented in order to carry out the evaluation for intertemporal relative efficiency by using the DEA moving split-windows analysis method. For the purpose of conducting an analysis of operational effectiveness, a complete sample of DMUs from a certain time period was employed as a reference set. The DEA moving split-windows had the window width set to a specific period of the DMU reference set for every moving window. The fundamental idea behind the DEA moving split-windows analysis is laid out in Table 6.

## 6. Case Demonstration

### 6.1. Empirical Data

A Chinese-context-based demonstration was conducted to demonstrate the application and effectiveness of the Ecological Well-being Performance (EWP) expressed in Section Four. Empirical data applied in the demonstration for the EWP evaluation were collected at provincial level in China for the period from 1997 to 2019.

There are a total of 34 provincial administrative districts in China at present. Hong Kong, Macau, Taiwan, and Tibet were excluded from the research due to the severe lack of data resources. As a result, 30 provinces in China were selected as area samples for empirical study. Considering that Chongqing was set as a municipality directly under the central government in 1997, relevant research data from 30 provinces were collected during the period from 1997 to 2019.

The investigation of the ecological well-being performance (EWP) among more than 30 provinces in China can provide a valuable reference for understanding whether various ecological resources and environmental pollutants have been delivered and utilized properly to generate human objective happiness to support sustainable urbanization and an ecological civilization. Therefore, the demonstration in the 30 sample provinces of China can not only show the effectiveness of the established EWP model but can also provide instrumental references and policy implications for city managers and urban planners in terms of urban ecological and environmental protection.

The data from 30 provinces in China was collected for the period from 1997 to 2019 in this empirical case study. The data sources regarding all indicators for the EWP evaluation in Table 3 are listed in Table 7.

### 6.2. Calculation Results

#### 6.2.1. Temporal Evolution

By applying the empirical data from indicators ECI-Xi and HDI-Yi among 30 sample provinces from 1997 to 2019 in China, the indicator values can be collected and obtained. Then, by applying the indicator values of 30 sample Chinese provinces from 1997 to 2019 to the EWP model described in Section Four, the values of the effectiveness assessment and efficiency assessment for the EWP among the 30 sample Chinese provinces during the period of 1997 to 2019 can be measured and ranked. Specifically, the values and rank of the ecological consumption index (ECI), human development index (HDI), and the efficiency of the EWP can be obtained and calculated, as is shown in Appendix A. By using the data in Appendix A, the changing trends of the average values of the ECI, HDI, and EWP efficiency among 30 provinces in China during the studied period can be captured and are shown in Figure 4a,b.

It can be seen from Figure 4a that both the ECI and the HDI have been increasing among the 30 provinces from 1997 to 2019, suggesting that residential objective happiness has been improving and the demands for ecological and environmental resources have also been increasing at the same time. Figure 4b shows a fluctuant, decreasing trend of the EWP efficiency, with the highest value of 0.892 occurring in 2000 and the lowest value of 0.762 occurring in 2015. Both the annual average values of the ECI and HDI among 30 provinces in China have increased, with fluctuations from 1997 to 2019. Surprisingly, the annual average value of the EWP efficiency among the 30 provinces in China has decreased, with fluctuations from 1997 to 2019.

#### 6.2.2. Top and Bottom Three Performers

According to Appendix A, the top three provinces with high values for the HCI are Ningxia (0.3184), Inner Mongolia (0.3022), and Qinghai (0.2705), and the bottom three cities are Hainan (0.0861), Yunnan (0.0925), and Henan (0.0944), as is shown in Figure 5a. According to Appendix A, the top three provinces with high values for the HDI are Shanghai (0.858), Beijing (0.846), and Tianjin (0.84), and the bottom three provinces with low HDI values are Guizhou (0.691), Qinghai (0.703), and Yunnan (0.707), as is shown in Figure 5b. In Appendix A it can be seen that the first three provinces with high efficiency values for the EWP are Hainan (1.0591), Henan (1.036) and Yunnan (1.0303), and the last three provinces are Ningxia (0.3367), Liaoning (0.4377), and Jilin (0.4937), as shown in Figure 5a–c.

As is shown in Figure 5, the average ECI value presented by Ningxia is noticeably the highest, followed individually by Inner Mongolia, and Qinghai. The reasons for which these three provinces need more ecological consumption input could be attributed to their geographic positions, the fragility of their natural environment, and poor ecological carrying capacity [93,94]. On the contrary, Hainan, Yunnan, and Henan demonstrated the lowest HCI values. Hainan and Yunnan have some common characteristics of ecological consumption in terms of their dominant geographic location and reasonable industrial structure [95,96]. Henan, perhaps as a major agricultural province in central China, has a weak foundation and distinct industrial advantages for pursuing excessive ecological-resource consumption [97,98].

As is shown in Figure 5b, Shanghai, Beijing, and Tianjin were the top three performers in terms of the HDI among the 30 provinces. The reasons for which these provinces ranked in the top three in improving human well-being could be attributed to their booming economies, better health-treatment services, and advantaged educational resources that focus on generating more opportunities to pursue high-HDI performance [99,100]. Guizhou, Yunnan, and Qinghai were the three provinces with the lowest HDI values (Figure 5b). There are some common characteristics amongst these provinces which have poor performance on the improvement of human well-being in terms of the HDI, including (1) a cold climate and severe geographic location and (2) an inadequate public health system and basic medical care.

In terms of the EWP efficiency, as is shown in Figure 5c, Hainan, Henan, and Yunnan were the top three performers with respect to improving the ecological well-being performance, respectively. Meanwhile, these three provinces enjoyed the lowest performance with respect to HCI values. Hainan had the highest EWP value, which may be due to its comparatively developed tertiary industry. Hainan has been concentrating on building a duty-free international tourism island and promoting the transformation and upgrade of its industrial structure [101]. Conversely, the average value of EWP efficiency presented by Ningxia was noticeably the lowest, followed individually by Liaoning and Jilin (Figure 5c). There are several prevalent causes for their poor performance [99]. Liaoning and Jilin are typical, resource-based industrial and mining regions in northeast China with lagging economies. The ecological environments in these two provinces were considered to be particularly vulnerable, resulting in a decreased EWP efficiency. As economically underdeveloped provinces, these two provinces have made fewer efforts to address education and medical difficulties, which have contributed to their poor environmental well-being record [60]. Ningxia, the worst EWP performer, has inadequate resources for carrying out ecological conservation measures due to its low economic development level. According to the research by Bian et al. [41], Ningxia is a typical polluted and resource-depleted province that serves as China’s major nonferrous metals industrial base; thus, its economic growth is heavily based on the resource industry.

### 6.3. Analysis of EWP Effectiveness

A two-axis, four-quadrant scatter plot was created by using the four-quadrant evaluation framework established in Figure 6, and the location of each quadrant is shown in Figure 7.

As is shown in Figure 6 and Figure 7, there are eleven provinces located in the first quadrant of the quadrantal diagram of EWP efficiency, most of which are located in central and southwest China. The levels of the ecological environment input and resident welfare output are both relatively high, and these provinces are in a less-sustainable development state. Despite the fact that the standard of living in the aforementioned provinces has achieved a high level, the ecological consumption may exceed the regional ecological carrying capacity threshold. In a sense, the residential high standard of living is a result of their excessive consumption of ecological resources and consequent environmental sacrifice. There are seven provinces located in the second quadrant of the quadrantal diagram of EWP efficiency. The primary characteristics of these “H-L”-type provinces are: a high level of ecological environment input and a low level of resident-welfare output, which is the least desirable development state. The majority of the aforementioned provinces are located in northwest China, and their objective geographical location contributes to challenges such as a low ecological carrying capacity and a delicate natural environment. In addition, the region’s economic basis is rather poor, its degree of openness to the outside world is low, its infrastructure is imperfect, and transportation is inconvenient. For a long time, these have contributed to a low standard of living in the aforementioned provinces, which necessitates an increase in human, material, and financial resources, as well as a substantial investment in ecological and environmental resources. There are seven provinces located in the third quadrant of the quadrantal diagram of EWP efficiency. The primary features of these “L-L”-type provinces are that the input levels of the ecological environment and the output levels of residents’ welfare are comparatively low, and that the ecological welfare level indicates a less-sustainable development stage. It can be noted that the majority of the provinces listed above are situated in southeast China. There are five provinces located in the second quadrant of the quadrantal diagram of EWP efficiency. The primary characteristics of these “L-H”-type provinces are: a low level of ecological environment input, a high level of resident-welfare output, and a sustained development of ecological welfare levels, which represents the optimum stage of development. It can be seen that the aforementioned provinces, while improving the standard of living of their citizens, are also better at controlling their input of resources and energy, as well as the level of environmental pollution emitted, thus achieving the coordinated development of the ecological environment and economic society.

### 6.4. Analysis of EWP Efficiency

#### 6.4.1. Overall Analysis

According to the studies, 30 sample provinces were divided into five groups based on their EWP efficiencies using the equal interval classification method [4,66]. The five classification groups are as follows (σ represents EWP efficiency): poor (σ < 0.4), fair (0.4 ≤ σ 0.6), good (0.6 ≤ σ < 0.8), better (0.8 ≤ σ < 1), and excellent (σ ≥ 1). Based on the data in Appendix A, the proportions in each classification can be categorized; they are displayed in Figure 8.

As is shown in Figure 8, it is worth noting that none of the provinces were in the poor group for EWP efficiency in 2009. However, the number of provinces in the poor group increased from 2010 to 2019. Proportions with a fair classification were maintained at approximately 20% during the study period. The number of provinces with a good classification showed an increase from 10% in 1997 to 17% in 2011 and have kept the performance since then. The number of cities with a good classification changed dramatically: increasing from 0 in 1997 to 30 in 2003, remaining around 30% for most years after 2003. As for excellent classification, the proportion rose from 57% in 1997 to 63% in 2003, and was approximately 40% during the period of 2003 to 2019, except for an instance of reaching 57% in 2016. It can be seen from Figure 8 that there is still a need for continuing progress in the field of EWP efficiency, especially given that the number of poorer provinces has been increasing and the number of excellent performers has been decreasing since 2014.

#### 6.4.2. Regional Comparative Analysis

This study presents the regional distribution of the EWP efficiency so that a comparative analysis may be made of the regional differences among the 30 provinces. The traditional method of four-region division (East, Middle and West, Southeast) in China was overgeneralized in classifying the 30 Chinese provinces [102]. The regional division method, “Eight Economic Region”, which was proposed by the Research Center of The State Council of China in 2004 (see Table 8), aims to ensure that the number of regions is moderate, their geographical locations are comparable, and their levels of economic and social development are comparable. Therefore, this regional classification was selected in order to analyze regional disparities in the ecological well-being performance, as is shown in Figure 8.

As is shown in Figure 9, the EWP efficiency ranking of the northwest region and the middle reaches of the Yellow River region generally remained in the top two during the entire study period, while the northern coastal area and the middle reaches of the Yangtze River generally belonged to the third and fourth places. The eastern and southern seas were, for a long time, in the seventh and eighth places. Further statistics show that the northwest region was in the first place for eighteen years, followed by the middle reaches of the Yellow River location, which was in the first place for five years. The southern coastal region was in the seventh place nine times and the eighth place eight times. It is a very interesting finding that the good performance areas were generally found at high levels and poor regions were found at low levels throughout the study period. In other words, the relative performance of the eight regions during the study period was relatively stable.

## 7. Discussion

### 7.1. Effectiveness of Proposed EWP Evaluation Method

The evaluation methodologies used in the previous study for the EWP do not differentiate between the effectiveness assessment and the efficiency assessment of the EWP. The evaluation approach that was developed for the purpose of analyzing the ecological well-being performance (EWP) in this study considers both the effectiveness and efficiency points of view, as well as the connections that exist between the two.

The indicator system for measuring the ecological well-being performance (EWP) was established by introducing an input–output model. This paper has introduced a matrix for evaluating the effectiveness of the EWP status by using the ECI and HDI, with a primary emphasis on the ecological consumption input and the residential well-being output. On this basis, an additional evaluation of the EWP efficiency was carried out by implementing a unified indication and making use of the input–output concept. The Ecological Consumption Index (ECI) and the Human Development Index (HDI) were selected to evaluate the EWP. The entropy method, line-weighted method, and four-quadrant evaluation framework were used to disclose the EWP effectiveness. The Super-SBM and DEA moving split-windows analysis method were applied to calculate the EWP efficiency. The evaluation indicators for the EWP in this study were made up of three types of well-being output in addition to six types of ecological input. To be more specific, the Ecological Consumption Index (ECI) is qualified by the six types of ecological input, whereas the Human Development Index (HDI) is qualified by the three types of well-being output. Meanwhile, the EWP efficiency was measured by the above six categories of ecological input and the three categories of well-being output. These indicators, which were established by this improved EWP evaluation method, were able to offer a comprehensive guidance for analyzing the urban ecological well-being performance (EWP) by incorporating assessments of effectiveness and efficiency into a unified evaluation framework. By using this improved method, the effectiveness of the EWP was classified into four levels, namely: High ECI–High HDI (H-H), High ECI–Low HDI (H-L), Low ECI–Low HDI (L-L), and Low ECI–High HDI (L-H). These levels can be used to describe the development effectiveness of the EWP in different urban regions. In the meantime, the efficiency of the EWP can provide more assistance in determining whether it is the urban ecological consumption input that makes a greater contribution to improving the human well-being output. This identification will provide valuable references for policy-making in order to further improve the EWP by adjusting the ECI and HDI for the purpose of attaining sustainable urban development.

The most important factor to determine how successfully the suggested EWP evaluation technique is implemented is the data quality of the indicators. As a result, the ease of access to the data as well as its correctness can help to ensure that the outcomes of the evaluation are more appropriate and effective. When examining the EWP for a particular urban region or set of places, the study period selected to use should be long enough to provide accurate results. This is another crucial aspect of utilizing this method. If the EWP evaluation is conducted over a longer research period and the changing trends of the two variables (effectiveness and efficiency) can be identified in an efficient manner, the end results will have a greater amount of significance. In addition, the method may be used to compare the EWP in each urban region by performing research on a collection of urban regions. The results of these comparisons can then be utilized to improve the poor urban regions that have a negative impact on the EWP.

### 7.2. Main Findings from Case Demonstration

The demonstration in Section Five shows the effectiveness of the proposed EWP evaluation model. In this study, some interesting findings from the case demonstration can be elaborated on, as follows.

From the overall change of the ECI and HDI values described in Section 6.2, the phenomenon suggests that both the input of urban ecological consumption and the output of human well-being are on the rise, but the EWP efficiency is far from SBM-efficient and is relatively low [41]. The rise of urban ecological consumption is related to China’s rapid urbanization process, and with the continuous advancement of urbanization, the energy consumption of China’s urban development is increasing. One study by Liu et al. [103] also pointed out that urbanization has led to an increase in China’s energy consumption. It can be seen that controlling energy consumption is a problem that must be faced by sustainable practice in China. The increase in human well-being is related to the Chinese government’s continuous investment in education and medical care. The research by Zhang et al. [4] argued that the human well-being of the Chinese government is increasing due to the government’s heightened concern.

From the regional changes in the EWP effectiveness detailed in Section 6.3, it can be seen that the fourth type of cities in the sustainable mode are mostly located in the southern provinces and Chongqing City. On one hand, these southern cities do not need heating in winter and their energy consumption is relatively low. On the other hand, the industrial structure of these cities is mostly of a tertiary industry, which makes industrial energy consumption lower [64,104]. As a city in the western region, Chongqing has a lower energy consumption and a level of higher human well-being. Previous studies show that, due to the abundant tourism resources, Chongqing’s ecological energy consumption is not high, while the life satisfaction of residents is comparable [105,106].

In terms of regional differences in the EWP efficiency detailed in Section 6.4, the performance of the southern coastal areas has been relatively good, followed by the southwest region, the eastern coastal area, the northern coastal area, the middle reaches of the Yangtze River, the middle reaches of the Yellow River, the northwest region and the northeast region. It is evident that the most developed regions, such as Beijing, Shanghai, and Guangdong, did not present the best EWP values. On one hand, China’s manner of economic development is expansive, resulting in an inefficient resource consumption over the past three decades [103]. In addition, the manner of widespread economic growth has generated numerous eco–environmental issues in urban areas, such as PM2.5 pollution [107]. On the other hand, the rapid urbanization caused unsustainable expansion in a number of provinces, resulting in a relative scarcity of resources per inhabitant [108]. In addition, low economic and technological levels led to excessive resource consumption, high pollution emissions, and low human well-being, resulting in poor EWP values [109].

## 8. Conclusions

An effective and proper approach to evaluating the ecological well-being performance (EWP) should be developed and employed. Most previous studies focused on the evaluation of the EWP efficiency without considering the effectiveness of the EWP, which may mislead the practice of improving the EWP. This paper proposed a bi-dimensional effectiveness and efficiency perspective evaluation of the EWP to sculpt sustainable performance.

Using an input–output paradigm, an indicator system for assessing the ecological well-being performance (EWP) was established. To evaluate the EWP, the Ecological Consumption Index and the Human Development Index were chosen. Utilizing the entropy approach, the line-weighted method, and the four-quadrant evaluation framework, the effectiveness of the EWP was revealed. EWP efficiency was calculated using Super-SBM and the DEA moving split-windows analysis. The purpose of this study was to investigate the evaluation of the EWP from the perspective of a bi-dimensional view of both its effectiveness and its efficiency. In the meantime, it was decided to use the same indication system for determining the EWP’s effectiveness as well as its efficiency. The evaluation findings for the effectiveness and efficiency of the EWP can therefore be evaluated side-by-side. With respect to research methods, the weighting for the ECI evaluation through the use of the entropy approach was confirmed, and subjective evaluation variables were eliminated to the greatest extent possible. When measuring the output of residential well-being, it was more persuasive to use indicators of the HDI, which are considered to be internationally authoritative. The effectiveness of the EWP status across China’s 30 provinces was demonstrated using a four-quadrant evaluation methodology, which was utilized to illustrate the data clearly. The static DEA model was the only one that was capable of evaluating the relative efficiency of several DMUs over the same time period, which made it impossible to compare the evaluation results across other time periods. When working with panel data, the DEA moving split-windows analysis is a method that has been proven effective for evaluating a dynamic DEA.

Data from 30 provinces in China for the period of 1997 to 2019 was collected for empirical study to demonstrate the effectiveness of the proposed method. The main findings of the case study are: (1) the ECI and HDI were increased during the study period, while the annual average value of the EWP efficiency among the 30 provinces in China has decreased with fluctuation; (2) provinces in southern China and Chongqing had low ECI levels and good performance with respect to the HDI; these are the most sustainable regions in China from an EWP effectiveness perspective, and (3) most developed regions, such as Beijing, Shanghai, and Guangdong, did not present the best EWP values.

This study evaluated the the ecological welfare performance in China from the perspectives of effectiveness and efficiency, theoretically improving the research on the EWP, and also promoting the development of the urban sustainability theory. From a practical point, this study provides an empirical study on the performance of 30 provinces over the past 20 years, which can provide a basis for understanding their performances in three aspects: energy consumption, human well-being, and the EWP, so as to formulate targeted sustainable-development strategies. The research limitation of this study is that the empirical data only extend up to 2019; and further improvement of the data can lead to empirical research in 2020, 2021, and 2022. With the development of multi-data, smaller units of empirical research can be carried out in the future. Future research can generalize the analytical framework of bi-dimensional effectiveness and efficiency perspectives to other fields.

## Figures and Tables

**Figure 1 ijerph-20-02024-f001:**
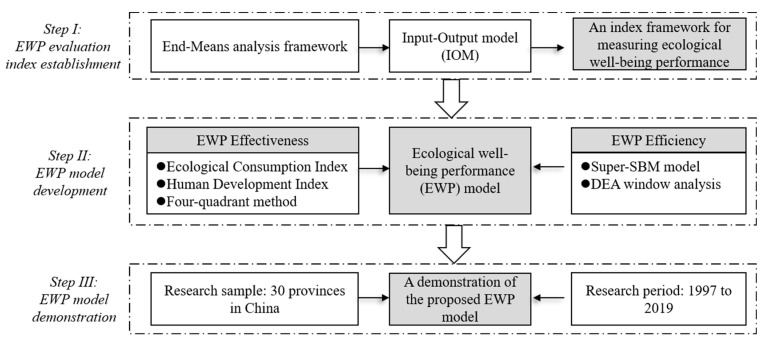
Research procedures designed in this study.

**Figure 2 ijerph-20-02024-f002:**
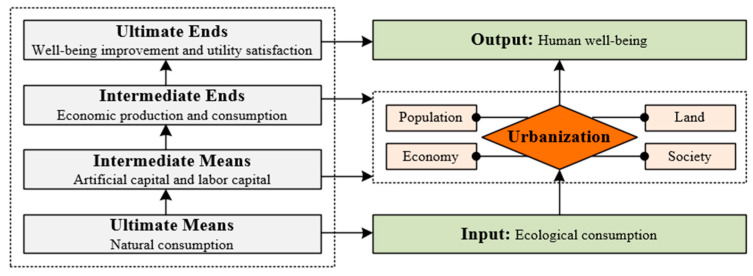
The input–output mode (IOM) for EWP evaluation.

**Figure 3 ijerph-20-02024-f003:**
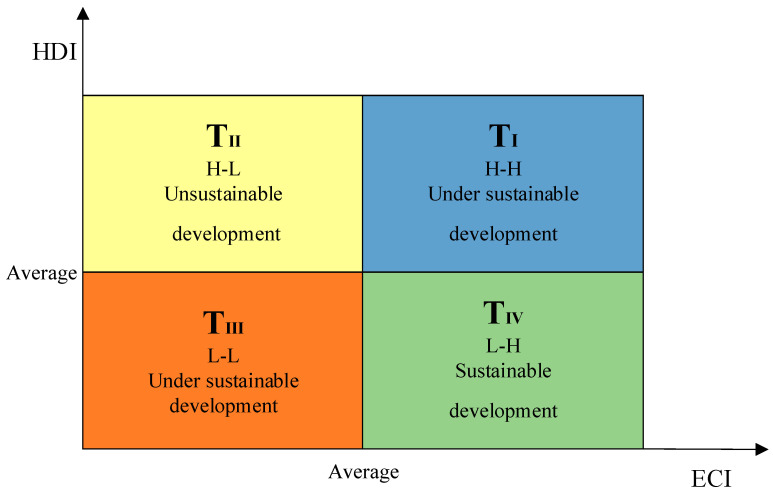
The four-quadrant evaluation framework for effectiveness assessment of EWP.

**Figure 4 ijerph-20-02024-f004:**
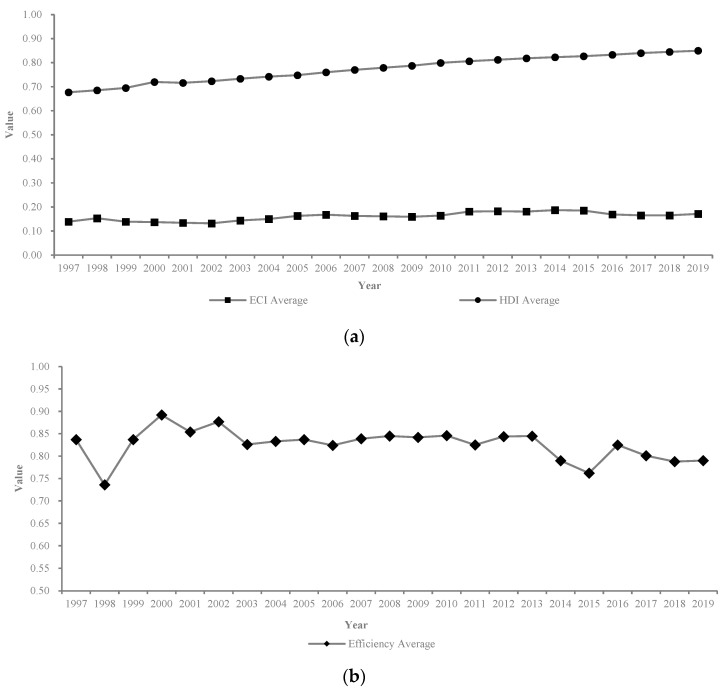
(**a**) The average values of the ECI and HDI among 30 provinces from 1997 to 2019 in China. (**b**) The average value of the EWP efficiency among 30 provinces from 1997 to 2019 in China.

**Figure 5 ijerph-20-02024-f005:**
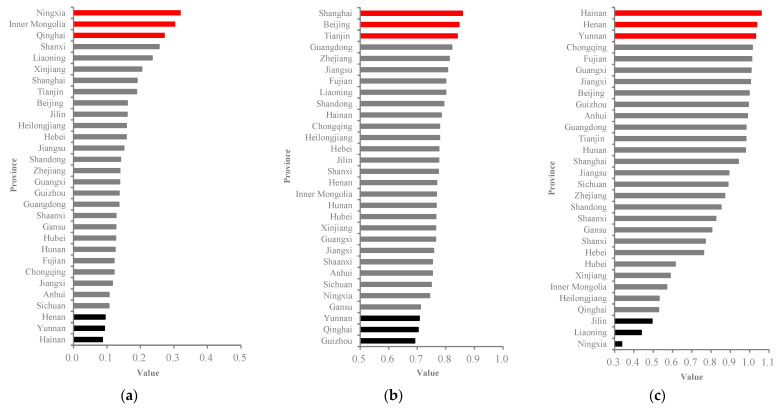
(**a**). The average provincial value of the ECI among 30 provinces from 1997 to 2019 in China. (**b**) The average provincial value of the HDI among 30 provinces from 1997 to 2019 in China. (**c**) The average provincial value of the EWP efficiency among 30 provinces from 1997 to 2019 in China.

**Figure 6 ijerph-20-02024-f006:**
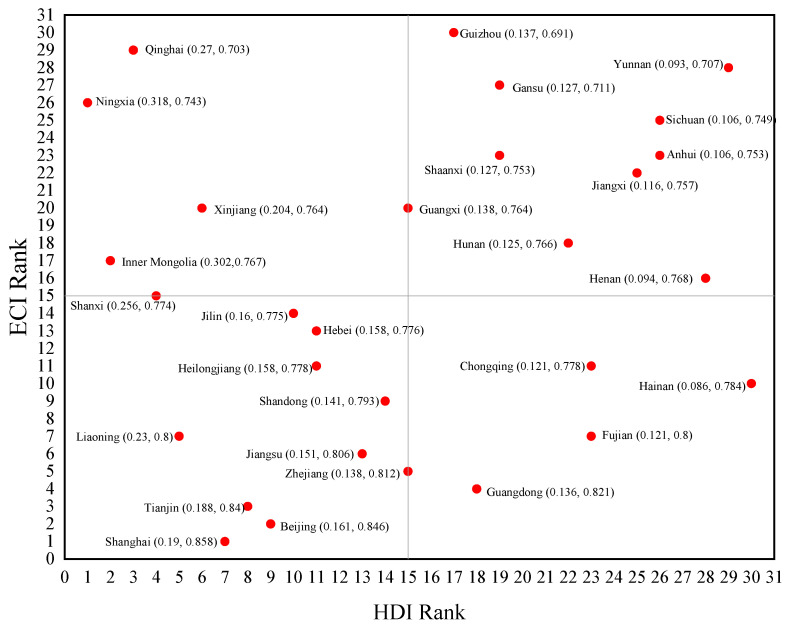
Quadrantal diagram of EWP effectiveness types of 30 provinces in China.

**Figure 7 ijerph-20-02024-f007:**
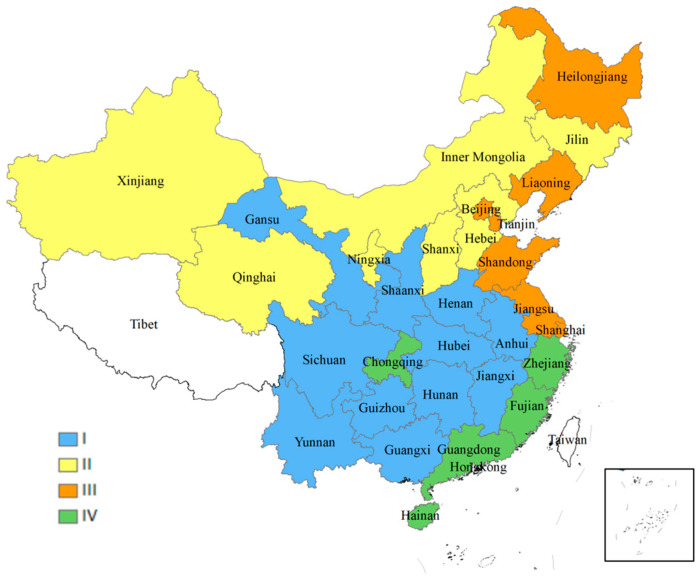
Locations of EWP effectiveness types of 30 provinces in China.

**Figure 8 ijerph-20-02024-f008:**
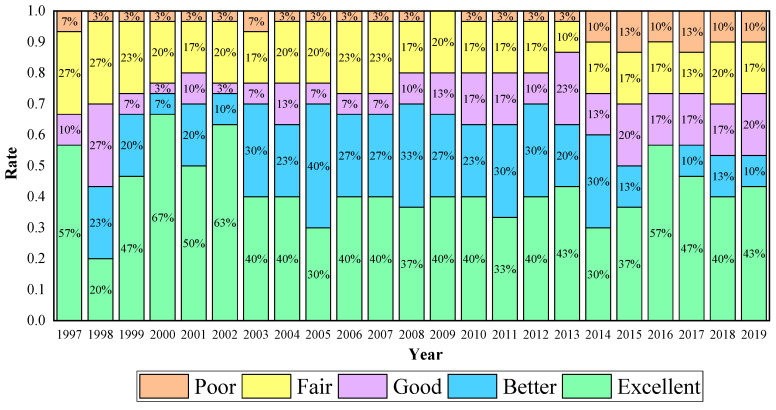
Proportions in each classification of EWP efficiency from 1997 to 2019.

**Figure 9 ijerph-20-02024-f009:**
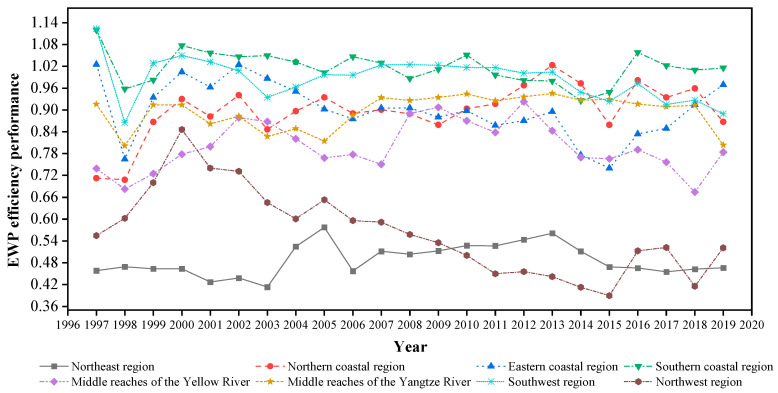
Regional comparative for the EWP efficiency performance.

**Table 1 ijerph-20-02024-t001:** Candidate indicators for evaluating ecological consumption.

Classification	Two-Level Index	Typical Indicator (Reference)
Resource consumption	Energy resource consumption	Per capita standard coal consumption; per capita electricity consumption; per unit energy consumption of GDP; per unit electricity consumption of GDP [11,47,48,49,50,51,52]
	Land resource consumption	Per capita built-up area; per capita arable land; per capita construction land [47,48,49,50,53]
	Water resource consumption	Per capita water supply [52,54]
Environment Pollution	Wastewater discharge	Per capita total discharge of industrial wastewater; per capita COD of industrial wastewater; per capita NH-N of industrial wastewater; per capita total discharge of domestic wastewater; per capita COD of domestic wastewater; per capita NH-N of domestic wastewater [48,50,55,56,57,58]
	Exhaust emissions	per capita SO_2_ emission from industrial waste gas; per capita industrial fumes emission; per capita industrial dust emission; per capita domestic fumes emission; per capita domestic dust emission; per capita CO_2_ emission [50,56,57,58,59]
	Municipal solid-waste discharge	Per capita industrial municipal solid-waste discharge; rate of multipurpose utilization; quantity of domestic garbage clearance disposal; rate of garbage innocuity disposal [53,60,61,62]

**Table 2 ijerph-20-02024-t002:** Typical indicators for evaluating well-being improvement.

Typical Indicator Recommended	Evaluation Content/Index	Year	Research Institution and Researcher (Reference)
Gross National Happiness (GNH)	Society development, traditional culture, ecological environment, and government management, etc.	1970	Jigme Singye Wang chuck [72]
Physical Quality of Life Index (PQLI)	Infant mortality rate, average life expectancy, literacy rate (above 15 years old)	1979	Overseas Development Committee (USA) [73]
Index of Social Progress (ISP)	Thirty-six indicators, including education, health, economy, population, geography, and culture, etc.	1984	International Council on Social Welfare (ICSW) [74]
Index of Sustainable Economic Welfare (ISEW)	Health, education, natural resources, ecological environment, and distributive justice, etc.	1989	Daly [75,76]
Human Develop Index (HDI)	Life expectancy, years of education, and standard of living	1990	The United Nations Development Programme (UNDP) [77]
Happy Planet Index (HPI)	Life satisfaction, carbon dioxide emissions, and average life expectancy	2006	New Economics Foundation (NEF) (UK) [78]
Your Better Life Index	Eleven categories, including housing, income, employment, education, environment, sanitation, community life, institutional management, security, harmony between work and family, living conditions, social contact, etc.	2011	Organization for Economic Co-operation and Development (OECD) [70]
World Happiness Index (WHI)	Nine fields, including education, health, environment, management, diversity and inclusiveness of culture, community vitality, subjective happiness, standard of living, etc.	2012	United Nations (UN) [79]

**Table 3 ijerph-20-02024-t003:** Main feature descriptions for the evaluation indicator system of EWP.

Dimension	Criteria	Two-Level Index	Specific Indicator (Unit)
Input indicator	Environmental Consumption Index (ECI)	Energy consumption	ECI-X_1_ Per capita standard coal consumption (tce)
		Land consumption	ECI-X_2_ Per capita built-up area (m^2^)
		Water consumption	ECI-X_3_ Per capita water supply (M^3^)
		Wastewater discharge	ECI-X_4_ Per capita COD of wastewater (kg)
		Exhaust emission	ECI-X_5_ Per capita SO_2_ emission (kg)
			ECI-X_6_ Per capita fumes emission (kg)
		Municipal solid-waste discharge	ECI-X_7_ Quantity of domestic garbage clearance disposal (t)
			ECI-X_8_ Per capita industrial solid waste discharge (t)
Output indicator	Human Development Index (HDI)	Economic development	HDI-Y_1_ Per capita GDP (Yuan)
		Residential health level	HDI-Y_2_ Life expectancy at birth (Year)
		Residential education level	HDI-Y_3_ Adult literacy rate (%)
			HDI-Y_4_ Comprehensive education enrollment rate (%)

Note: The data of GDP are normalized to the constant prices of 1997 in order to avoid the impact of economic inflation.

**Table 4 ijerph-20-02024-t004:** The index benchmark of Human Development Index (HDI).

Index (Unit)	Minimum	Maximum
HDI-Y_1_ Per capita GDP (Dollar)	100	40,000
HDI-Y_2_ Life expectancy at birth (Year)	25	85
HDI-Y_3_ Adult literacy rate (%)	0	100
HDI-Y_4_ Comprehensive education enrollment rate (%)	0	100

Reference: Human Development Report 2021/2022 [80,81].

**Table 5 ijerph-20-02024-t005:** The evaluation classification for effectiveness assessment of EWP.

Type	Classification	Characteristic	Effectiveness Performance
Type I	C_H-H_	High ECI-High HDI	Under sustainable development
Type II	C_H-L_	High ECI- Low HDI	Unsustainable development
Type III	C_L-L_	Low ECI-Low HDI	Under sustainable development
Type IV	C_L-H_	Low ECI-High HDI	Sustainable development

**Table 6 ijerph-20-02024-t006:** The basic principle of DEA moving split-windows analysis.

		T_1_	T_2_	…	T_d_	T_d+1_	…	T_m-d_	T_m-d+1_	…	T_m-1_	T_m_
DMU_1_	Window_1_	δ_111_	δ_112_	…	δ_11d_							
	Window_2_		δ_121_	…	δ_1,2,d−1_	δ_12d_						
	…						…					
	Window_m-d_							δ_1,m−d,1_	δ_1,m−d,2_	…	δ_1,m−d,d_	
	Window_m-d+1_								δ_1,m−d+1,1_	…	δ_1,m−d+1,d-1_	δ_1,m−d+1,d_
	Mean value											
DMU_2_	Window_1_	δ_211_	δ_212_	…	δ_21d_							
	Window_2_		δ_221_	…	δ_2,2,d−1_	δ_2,2,d_						
	…						…					
	Window_m-d_							δ_2,m−d,1_	δ_2,m−d,2_	…	δ_2,m−d,d_	
	Window_m-d+1_								δ_2,m−d+1,1_	…	δ_2,m−d+1,d-1_	δ_2,m−d+1,d_
	Mean value											
…												
DMU_n_	Window_1_	δ_n11_	δ_n12_	…	δ_n1d_							
	Window_2_		δ_n21_	…	δ_n,2,d−1_	δ_n2d_						
	…						…					
	Window_m-d_							δ_n,m−d,1_	δ_n,m−d,2_	…	δ_n,m−d,d_	
	Window_m-d+1_								δ_n,m−d+1,1_	…	δ_n,m−d+1,d-1_	δ_n,m−d+1,d_
	Mean value											

**Table 7 ijerph-20-02024-t007:** Data sources of all indicators for the EWP evaluation.

Indicator	Data Sources
HDI-Y_1_	China’s Statistical Yearbooks (1998–2020)
ECI-X_1_	China’s Energy Statistical Yearbooks (1998–2020)
HDI-Y_2_	China Economic and Social Development Statistical Database China’s Population and Employment Statistical Yearbook (1998–2020) China Population Census (1990/2000/2010/2020)
HDI-Y_3_HDI-Y_4_	China’s Education Statistical Yearbook (1998–2020) Statistical Bulletin of Education Development (1998–2020) China Economic and Social Development Statistical Database
ECI-X_2-8_	China’s Environmental Statistical Yearbook (2004–2020) Environmental Statistics Bulletin (1998–2020) Statistical Bulletin of National Economic and Social Development (1998–2020) Statistical Database from China Economic Information Network

**Table 8 ijerph-20-02024-t008:** The classification of China using the “Eight Economic Region” method.

Region	Provinces
Northeast region	Liaoning; Jilin; Heilongjiang
Northern coastal region	Beijing; Tianjin; Hebei; Shandong
Eastern coastal region	Shanghai; Jiangsu; Zhejiang
Southern coastal region	Fujian; Guangdong; Hainan
Middle reaches of the Yellow River	Inner Mongolia; Shaanxi; Shanxi; Henan
Middle reaches of the Yangtze River	Hubei; Hunan; Jiangxi; Anhui
Southwest region	Sichuan; Chongqing; Yunnan; Guizhou; Guangxi
Northwest region	Ningxia; Xinjiang

## Data Availability

The data presented in this study are available on request from the corresponding author.

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
