# Peer review of "A Effectiveness-and Efficiency-Based Improved Approach for Measuring Ecological Well-Being Performance in China"

_ijerph, 2023, doi:10.3390/ijerph20032024_

Round 1

Reviewer 1 Report

A review on the manuscript in International Journal of Environmental Research and Public Health entitled „An effectiveness and efficiency bidimensional based improved approach for measuring ecological well-being performance in China“

This paper proposes a two-dimensional prospective evaluation of EWP effectiveness and efficiency for sustainable performance design. The system of ecological well-being performance measurement indices (EWP) is created by introducing the input-output model. Ecological consumption index and human development index are selected to evaluate EWP. Entropy method, line-weighted method and four-quadrant evaluation framework are used to evaluate EWP efficiency.

Broad comments

The methods used to assess the effectiveness and efficiency of sustainable performance are described in depth. The proposed methods have been evaluated on the basis of an empirical dataset collected for 30 Chinese provinces from 1997 to 2019. The results and the resulting conclusions are based on the analysis and are sufficient.

The article needs minor technical correction.

Specific comments

Formulas of chemical compounds must be formatted correctly using indices - instead of "CO2" (lines 2023, 238) it should be "CO2" ; "SO2" (lines 238, 372) should be "SO2".

Figures consisting of several parts must be marked separately, for example (a), (b), (c) and the corresponding explanations must be provided in the caption of the figure, not in the graphic area of the figure. Figure 4 , Figure 5 and Figure 6. Each figure can have only one caption (Figure 6). It is recommended to read Figure 6 (a) and Figure 6 (b) as different figures, as their content is completely different.

The charts lack axis titles (Figures 4, 7, 8 - on vertical axes; Figure 5 - on horizontal axes).

Equations numbers must all be right-aligned, the current alignment is uneven anyway.

Author Response

No

Comment

Response

0

This paper proposes a two-dimensional prospective evaluation of EWP effectiveness and efficiency for sustainable performance design. The system of ecological well-being performance measurement indices (EWP) is created by introducing the input-output model. Ecological consumption index and human development index are selected to evaluate EWP. Entropy method, line-weighted method and four-quadrant evaluation framework are used to evaluate EWP efficiency.

Thanks for the reviewer’s very supportive comments.

We have studied all the reviewers’ comments carefully and have made corrections and improvements accordingly.

1

Formulas of chemical compounds must be formatted correctly using indices - instead of "CO2" (lines 2023, 238) it should be "CO2"; "SO2" (lines 238, 372) should be "SO2".

Thanks for the reviewer’s very kind comments.

The authors apologize for the clerical errors. All formulas of chemical compounds in paper have been checked and formatted correctly. All formulas of "CO2" and "SO2" have been already revised into “CO2” and “SO2”.

2

Figures consisting of several parts must be marked separately, for example (a), (b), (c) and the corresponding explanations must be provided in the caption of the figure, not in the graphic area of the figure. Figure 4, Figure 5 and Figure 6. Each figure can have only one caption (Figure 6). It is recommended to read Figure 6 (a) and Figure 6 (b) as different figures, as their content is completely different.

Thanks to the reviewer’s useful suggestions.

Figure 4 have been already marked and renamed new captions as Figure 4(a) and Figure 4(b) separately. Figure 5 have been already marked and renamed new captions as Figure 5(a), Figure 5(b), and Figure 5(c) separately. Figure 6 (a) and Figure 6 (b) have already marked and renamed new captions as Figure 6 and Figure 7 separately.

3

The charts lack axis titles (Figures 4, 7, 8 - on vertical axes; Figure 5 - on horizontal axes).

Thanks for the reviewer’s very kind comments.

The authors apologize for the technical mistakes in drawing charts. All axis titles of charts in paper have been checked and revised. The vertical axes and the horizontal axes of charts for Figure 4, 5,7, and 8 have already been added.

4

Equations numbers must all be right-aligned, the current alignment is uneven anyway.

Thanks for the reviewer’s very kind comments.

The authors apologize for the technical mistakes in equations numbers format. All equations numbers have already revised right-aligned.

Reviewer 2 Report

The paper contains interesting results of the study on relations between economic and environmental aspects of wellbeing. The paper is well written, English is good.

The paper fulfills the standards of a scientific article, however there are the elements which should be improved. In the introduction the authors must better explain in what way they wanted to evaluate the effectiveness and the effectiveness. Beside they should write what is the meaning of these two terms according to them. What does the effectiveness of EWP mean and the effectiveness mean? In the section 4 “methodology” they should connect these explanations with the methods they used in the research.

The title of the section 4.3 “Indicator selection of human development index (HDI)” is wrong. It suggests that the authors wanted to choose one of the “human development indexes”. In fact they have chosen one of presented measures of economic and social welfare. The choice is the HDI.

In the discussion the authors wrote that: “The evaluation approach that was developed for the purpose of analyzing the ecological well-being performance (EWP) in this study takes into account both the effectiveness and efficiency points of view, as well as the connections that exist between the two.” They should clearly explain the results on these “connections”. In the further part they wrote that: “EWP efficiency is measured by the above six categories of ecological input and three categories of well-being output”. They did not write how they measured the effectiveness of EWP in that section (something about it is in the conclusion). They should do it. They wrote that “the effectiveness of EWP is classified into four levels, namely, High ECI-High HDI (H-H), High ECI- Low HDI (H- 770 L), Low ECI-Low HDI (L-L), and Low ECI-High HDI (L-H)”. That is the result of the measurement but not the method of measurement.

In the conclusion the authors should explain the role of Entropy method, weighted method, and Four-quadrant evaluation framework in the development of evaluation of the EWP effectiveness. The same about the Super SBM and DEA Moving Split-windows Analysis method which were applied to calculate EWP efficiency.

Author Response

No

Comment

Response

0

The paper contains interesting results of the study on relations between economic and environmental aspects of wellbeing. The paper is well written, English is good.

Thanks for the reviewer’s very supportive comments.

We have studied all the reviewers’ comments carefully and have made corrections and improvements accordingly.

1

The paper fulfills the standards of a scientific article, however there are the elements which should be improved. In the introduction the authors must better explain in what way they wanted to evaluate the effectiveness and the effectiveness. Beside they should write what is the meaning of these two terms according to them. What does the effectiveness of EWP mean and the effectiveness mean? In the section 4 “methodology” they should connect these explanations with the methods they used in the research.

Thanks to the reviewer’s useful suggestions.

In Introduction Section, the specific definitions and meanings for “effectiveness” and “efficiency” of EWP have been reemphasized in paper (shown in red font with yellow paragraph from Line127 to 136). More descriptions were added.

In Methodology Section, Figure 1 and relevant content show and define the explanations for “effectiveness” and “efficiency” of EWP with the methods. The paper tries to build up a bidimensional model to comprehensively measure ecological well-being performance from the two angles of effectiveness and efficiency.

2

The title of the section 4.3 “Indicator selection of human development index (HDI)” is wrong. It suggests that the authors wanted to choose one of the “human development indexes”. In fact they have chosen one of presented measures of economic and social welfare. The choice is the HDI.

Thanks to the reviewer’s useful suggestions.

The title of Section “Indicator selection of human development index (HDI)” has been revised as “Indicator selection of human well-being index” (seen in Line 347).

3

In the discussion the authors wrote that: “The evaluation approach that was developed for the purpose of analyzing the ecological well-being performance (EWP) in this study takes into account both the effectiveness and efficiency points of view, as well as the connections that exist between the two.” They should clearly explain the results on these “connections”. In the further part they wrote that: “EWP efficiency is measured by the above six categories of ecological input and three categories of well-being output”. They did not write how they measured the effectiveness of EWP in that section (something about it is in the conclusion). They should do it. They wrote that “the effectiveness of EWP is classified into four levels, namely, High ECI-High HDI (H-H), High ECI- Low HDI (H- 770 L), Low ECI-Low HDI (L-L), and Low ECI-High HDI (L-H)”. That is the result of the measurement but not the method of measurement.

Thanks to the reviewer’s constructive comment.

In Discussion Section, the connection for “effectiveness” and “efficiency” of EWP has been revised and discussed. And the discussion for evaluation method for “effectiveness” and “efficiency” of EWP has been added. The updated details can be seen in red font with yellow paragraph from Line 751 to 760.

4

In the conclusion the authors should explain the role of Entropy method, weighted method, and Four-quadrant evaluation framework in the development of evaluation of the EWP effectiveness. The same about the Super SBM and DEA Moving Split-windows Analysis method which were applied to calculate EWP efficiency.

Thanks to the reviewer’s constructive comment.

In Conclusion Section, the vital role and importance of evaluated method, such as Entropy method, weighted method, Four-quadrant evaluation framework, Super SBM, and DEA Moving Split-windows Analysis method, have been revised and demonstrated. The updated details can be seen in red font with yellow paragraph from Line 834 to 853.
